# Changes in Artery Diameters and Fetal Growth in Cases of Isolated Single Umbilical Artery

**DOI:** 10.3390/diagnostics13030571

**Published:** 2023-02-03

**Authors:** Elena Contro, Laura Larcher, Jacopo Lenzi, Marina Valeriani, Antonio Farina, Eric Jauniaux

**Affiliations:** 1Division of Obstetrics and Prenatal Medicine, Department of Medicine and Surgery, IRCCS Sant’Orsola-Malpighi, University of Bologna, 40138 Bologna, Italy; 2Section of Hygiene, Public Health and Medical Statistics, Department of Biomedical and Neuromotor Sciences, Alma Mater Studiorum, University of Bologna, 40138 Bologna, Italy; 3EGA Institute for Women’s Health, Faculty of Population Health Sciences, University College London, Stanmore HA7 4LP, UK

**Keywords:** single umbilical artery, SUA, fetal growth restriction, FGR, low birth weight, umbilical artery diameter

## Abstract

Background—There are conflicting data in the international literature on the risks of abnormal fetal growth in fetuses presenting an isolated single umbilical artery (SUA), and the pathophysiology of this complication is poorly understood. Objective—To evaluate if changes in diameter of the remaining umbilical artery in fetuses presenting an isolated SUA are associated with different fetal growth patterns. Study design—This was a two-center prospective longitudinal observational study including 164 fetuses diagnosed with a SUA at the 20–22-week detailed ultrasound examination and 200 control fetuses with a three-vessel cord. In all cases, the diameters of the cord vessels were measured in a transverse view of the central portion of the umbilical cord, and the number of cord vessels was confirmed at delivery. Logistic regression and nonparametric receiver operating characteristic (ROC) analysis were carried out to evaluate the association of the umbilical artery diameter in a single artery with small for-gestational age (SGA) and with fetal growth restriction (FGR). The impact of artery dimension was adjusted for maternal BMI, parity, ethnicity, side of the remaining umbilical artery and umbilical resistance index (RI) in the regression model. Results—A significantly (*p* < 0.001) larger mean diameter was found for the remaining artery in fetuses with SUA compared with controls (3.0 ± 0.9 vs. 2.5 ± 0.6 mm). After controlling for BMI and parity, we found no difference in umbilical resistance and side of the remaining umbilical artery between the SUA and control groups. A remaining umbilical artery diameter of >3.1 mm was found to be associated with a lower risk of FGR, but this association failed to be statistical significant (OR = 0.60, 95% CI = 0.33–1.09, *p* value = 0.089). We also found that the mean vein-to-artery area ratio was significantly (*p* < 0.001) increased in the SUA group as compared with the controls (2.4 ± 1.8 vs. 1.8 ± 0.9; mean difference = 0.6; Cohen’s d = 0.46). Conclusion—In most fetuses with isolate SUA, the remaining artery diameter at 20-22 weeks is significantly larger than in controls. When there are no changes in the diameter and, in particular, if it remains <3.1 mm, the risk of abnormal fetal growth is higher, and measurements of the diameter of the remaining artery could be used to identify fetuses at risk of FGR later in pregnancy.

## 1. Introduction

The mature umbilical cord consists of an outer covering of flattened amniotic epithelial cells and an interior mass of loose connective tissue containing the umbilical vessels (Wharton’s jelly) [1]. These vessels derive from the allantoic duct, which initially carries two umbilical arteries and two umbilical veins. The right umbilical vein disappears, but the two umbilical arteries normally remain. In 0.5–2% of pregnancies [2,3], one of the umbilical arteries may not develop or regresses progressively during the first trimester. A single umbilical artery (SUA) cord is the most frequent anomaly in humans, and it is often found in syndromes such as aneuploidies, acardiac fetuses or sinrenomelia [4,5,6,7,8]. Major fetal anatomical defects are largely responsible for the high fetal and neonatal loss from this pathology and can affect any organ system.

Around two-thirds of fetuses presenting with a SUA have no other anomalies and are referred to as isolated SUA [9]. A higher incidence of fetal growth restriction (FGR) has been reported among fetuses with a SUA and may be present without any other congenital anomalies in 10 to 15% of the cases [4,10,11,12]. The pathophysiology of the slow fetal growth in fetuses with isolated SUA is poorly understood. It is assumed that changes in umbilical cord blood flow hemodynamics could have an impact of volume flow distribution within the umbilico-placental circulation. We have recently shown that umbilical Doppler pulsatility index (PI) values are on average 20% lower in cases of isolated SUA compared to normal three-vessel cord controls and that this difference remains constant between 23 and 40 weeks of gestation [13]. This finding indicated a change in the resistance to flow in the remaining arteries in SUA and highlighted the need to adapt umbilical artery indexes’ reference ranges to manage fetuses with SUA and abnormal fetal growth. 

The aim of the present study was to further investigate the impact of a SUA on fetal growth and, in particular, if changes in the diameter of the remaining artery could explain different growth patterns in fetuses with SUA.

## 2. Materials and Methods

### 2.1. Setting and Study Design 

We conducted a prospective longitudinal observational study at two centers (IRCCS Sant’Orsola Malpighi, University of Bologna, Italy, and Institute for Women’s Health, University College of London, United Kingdom). In both units, patient demographics data, previous obstetric history, ultrasound data and images and symptoms at the time of the first examination were recorded and stored in a specialized database (Viewpoint Version 5, Bildverargeritung GmbH, Munich, Germany). Pregnancies were dated according to the last menstrual period (LMP) and confirmed by crown-rump length (CRL) measurements at 11–14-week nuchal translucency scan.

The patients were managed according to their local unit protocol. Pregnancy and delivery data were collected from hospital records. Local institutional ethical committee approval was obtained by the principal investigator prior to the start of this study (UK NHS Health Research Authority, no. 18/WM/0328). The protocol and a waiver of consent were granted a favorable opinion, as all ultrasound records were examined within the center, and basic clinical data were collected using a standard clinical audit protocol. 

### 2.2. Patients and Ultrasound Examination

Patients diagnosed with a SUA included in the present study were recruited from a cohort of pregnant women attending the ultrasound units over a 7-year period ending June 2021. Examining of the umbilical cord including the evaluation of the number of cord vessels is included in our routine ultrasound protocols at the mid-trimester detailed ultrasound. In all cases, when SUA is suspected on gray-scale imaging, a color Doppler mapping of the placental and fetal insertions of the umbilical cord is performed. All ultrasound examinations are carried out transabdominally by experienced operators using high-resolution ultrasound equipment (Voluson 730 and E8 Expert, GE Medical Systems, Milwaukee, WI, USA). Multiple pregnancies, fetuses presenting with aneuploidy or other congenital anomalies and cases with active maternal smoking were excluded from the study group. The control group consisted of uncomplicated pregnancies in non-smoking mothers with a singleton fetus with no anomalies on ultrasound examination confirmed at birth collected during the same study period. In both groups, the number of cord vessels was confirmed at delivery.

The diameters of the cord vessels in both groups were measured in a magnified transverse view of the umbilical cord at the level of the median portion of the cord (Figure 1, Figure 2 and Figure 3) by two operators (EC and EJ) at the 20–22-week scan. In all cases, gestational age was calculated from the patient’s last menstrual period (LMP) and was confirmed by measurements of the fetal crown-rump length (CRL) at the 11–14-week ultrasound examination. Pediatric follow-up data were obtained from the neonatal clinical records at birth and routine post-natal visits. The primary outcome was the birthweight, and the secondary outcome was impact on fetal growth during pregnancy as evaluated by the ultrasound estimated fetal-weight (EFW) and the corresponding percentiles using the Hadlock regression formulae [14].

### 2.3. Statistical Analysis

All variables are presented as counts and percentages. Crude differences in demographic and clinical characteristics across the three study subgroups: normal growth (10–90th centile), SGA (defined as a birth weight of <than 10th percentile for gestational age) [15] and FGR (fetuses with an estimated fetal weight or abdominal circumference that is <than the 10th percentile for gestational age) [15] were evaluated with the Fisher’s exact test. 

Logistic regression and nonparametric receiver operating characteristic (ROC) analysis was carried out to investigate the association of umbilical artery diameter and vein-to-artery (V/A) area ratio in single-artery patients with FGR and SGA at delivery. More specifically, effect sizes were expressed as odds ratios (ORs), while the optimal cutoff values for artery diameter were determined using the Youden method, which maximizes the sum of the sensitivity and specificity; results were coincident when we used the Liu method, which maximizes the product of the sensitivity and specificity. Afterward, the impact of artery dimension was adjusted for maternal BMI, parity, umbilical artery position, and umbilical resistance index (RI) by including these variables as covariates in the regression model. 

In the secondary analysis, differences in umbilical artery diameter and V/A area ratio between cases (single artery) and controls (two arteries) were evaluated with two-tailed *t* test and were visualized with the aid of box plots (Figure 4). In the control group, diameter estimates were obtained as the average of the diameters of the two umbilical arteries, while area estimates contributing to the V/A ratios were obtained as the sum of the areas of the two umbilical arteries.

All analyses were carried out using Stata software, version 17 (StataCorp, 2021. Stata Statistical Software: Release 17. College Station, TX, USA: StataCorp LLC). The significance level was set at 5%, and missing data were treated as an additional category of variables, where necessary.

## 3. Results

A total of 164 consecutive cases were included in the study group. As shown in Table 1, the maternal ethnics distribution was as follow: Caucasian (90.2%), 12 were of African descent (7.3%), and 3 were of Asian descent (1.8%). The mean age was 33.2 ± 5.0 years, and mean BMI at 12 weeks was 25.7 ± 4.7 kg/m^2^. 

Table 2 presents the results of logistic regression and nonparametric receiver operating characteristic (ROC) analysis conducted on umbilical artery diameter and vein-to-artery (V/A) area ratio to identify SGA fetuses and cases of FGR. After controlling for BMI, parity, umbilical resistance and missing umbilical artery side, results were very similar to those of the unadjusted analysis (SGA: OR = 0.60, 95% CI = 0.24–1.49; IUGR: OR = 0.54, 95% CI = 0.28–1.03). 

The diameter of remaining umbilical artery was significantly (*p* < 0.001) increased in fetuses with SUA compared to controls (3.0 ± 0.9 vs. 2.5 ± 0.6 mm; mean difference = 0.4 mm; Cohen’s d = 0.62). There was also a significant (*p* < 0.001) increase in the mean V/A area ratio in the SUA group compared with the controls (2.4 ± 1.8 vs. 1.8 ± 0.9; mean difference = 0.6; Cohen’s d = 0.46). 

A diameter of >3.3 mm for the diameter of the single umbilical artery was not found to be associated with a significantly lower risk of SGA (OR = 0.72, 95% CI = 0.31–1.68, *p* value = 0.443). A diameter of >3.1 mm was found to be associated with a lower risk of FGR, but this association failed to be statistically significant (OR = 0.60, 95% CI = 0.33–1.09, *p* value = 0.089). After controlling for BMI, parity, umbilical resistance and umbilical artery position, results were very similar to those of the unadjusted analysis, which means that diameter showed a borderline significant association with FGR (SGA: OR = 0.60, 95% CI = 0.24–1.49, *p* value = 0.272; FGR: OR = 0.54, 95% CI = 0.28–1.03, *p* value = 0.063). 

The incidence of SUA/FGR was increased (20.0% vs. 10.9%) when the remaining umbilical artery was the right one, although this difference did not reach statistical significance (*p* value = 0.226). In particular, the incidence of SUA was 5.0% on the right side and 4.7% on the left side, while the incidence of FGR was 15.0% on the right side and 6.3% on the left side. 

## 4. Discussion

### 4.1. Principal Findings of the Study 

Our study adds to a previous study that around 16% of fetuses presenting an isolated SUA at the mid-pregnancy detailed fetal anatomy ultrasound examination will show abnormal growth later in pregnancy or at birth and identifies a relationship between changes in the diameter of the single umbilical artery and fetal growth. A smaller diameter of the remaining umbilical arteries at 20–22 weeks was associated with a higher incidence of FGR, suggesting that in those cases, the mechanism of slow fetal growth could be due to a failure of the remaining artery to compensate hemodynamically and could be used to identify fetuses at risk of FGR later in pregnancy.

### 4.2. Comparison with Existing Literature 

Our knowledge of the vascular mechanisms leading to abnormal fetal growth are almost exclusively derived from studies of fetuses with an umbilical cord containing two arteries and a vein. Giles et al. were the first to correlate fetal umbilical artery flow velocity Doppler waveforms with placental villi microvascular anatomy [16]. They correlated the blood flow resistance in the umbilical circulation obtained with Doppler ultrasound with the numbers of small muscular arterioles and tertiary stem villi. They found that the number of small arterial vessels was lower in pregnancies with a high resistance to blood flow and was associated with a higher incidence of FGR than in normal controls or in pregnancies with clinically suspected FGR with normal Doppler features. In isolated SUA, there is no evidence of associated anomalies of the villous circulation, suggesting that the abnormal fetal growth is due to hemodynamic changes in the entire fetal circulation when the remaining artery does not compensate for the loss of the other. 

We have previously demonstrated that normal reference ranges for umbilical PI are inadequate for fetuses with SUA [13]. Overall, the umbilical pulsatility index (PI) is 20% lower in fetuses with isolated SUA than in normal fetuses; however, in that study, we did not evaluate fetal growth patterns, and we did not measure the diameter of the single umbilical artery [13]. During the second half of pregnancy, the vascular network with the lowest resistance in the entire fetal circulatory system is at the level of the placental vascular bed [17,18]. According to Poiseuille’s law, a decrease in blood viscosity and/or in the vascular system length and an increase in the mean radius of the vascular system collectively decrease the resistance to flow [19]. The latter effect is particularly strong, with the resistance depending on the radius to the fourth power. In the present study, we found changes in the diameter of the remaining umbilical artery in fetuses with SUA compared to controls, supporting the concept of a general hemodynamic impact from early pregnancy in the subgroups of fetuses that are diagnosed with FRG later in pregnancy. 

Li et al. analyzed 77 fetuses with SUA, and according to our results, they found a higher umbilical artery area in the SUA group compared with the control group. They also found that in the isolated SUA group, the umbilical vein area/umbilical artery area ratio was lower than in the control group [20].

### 4.3. Clinical Implications

A population-based, retrospective cohort study of 37,500 singleton pregnancies including 223 SUA diagnosed at birth has found a higher incidence of birth weight < 10th percentile (OR 2.1; CI 1.44–2.93) in isolated SUA [21]. Similarly, a retrospective case-control series of 136 SUA diagnosed at second-trimester ultrasound has reported isolated SUA to be an independent risk factor for fetal growth restriction (adjusted OR = 11.3, 95% CI 4.8–25.6) compared to a normal three-vessel cord [22]. Two recent systematic reviews have reported odds ratios (OR) for SGA ranging between 1.6 (95% CI, 0.97–2.6) [10] and 2.75 (95% CI, 1.97 to 3.83) [23] in isolated SUA compared to controls with a normal umbilical cord. A meta-analysis of 11 observational studies including 1731 pregnancies had found a 2.75-fold risk of SGA in fetuses presenting with isolated SUA [23]. Further studies have also demonstrated an association between SUA and the risk of FGR stillbirths [24,25]. Finally, a retrospective analysis of over 200,000 pregnancies, excluding those presenting with structural or chromosomal anomalies, reported an OR of 8.1 for stillbirth with isolated single umbilical artery [26]. None of the above studies have evaluated the changes in fetal growth in relation with the change in the diameter of the remaining artery. The present study confirms an association between isolated SUA and abnormal fetal growth and reports that a cut-off of 3.1 mm for the diameter of the single umbilical artery at the 19–23-week detailed fetal anatomy ultrasound examination is associated with a borderline significant greater risk of FRG. These data suggest that measurements of the diameter of the remaining artery in fetuses with isolated SUA at mid-pregnancy could be used to identify fetuses at risk of abnormal growth later in pregnancy.

### 4.4. Strengths and Limitations of the Study 

Our study has a number of strengths compared to other published studies on SUA and fetal growth. As far as we know, this is the first study to have measured the diameters of umbilical vessels in relation to abnormal fetal growth in isolated SUA. This measurement is more simple and less time-consuming to make, compared to the area that was used in a recent study [20]. The comparison with controls allowed us to evaluate the impact of changes in diameter of the remaining artery and fetal growth from mid-pregnancy until birth. 

Another strength of our study, compared to a previously published study, is the relatively large number of cases in each group that has enabled the data to be adjusted for maternal BMI, parity and ethnicity in the regression model, thus controlling for the main factors affecting fetal growth.

The main weakness of our study rests in its retrospective nature, although this was mitigated by the fact that all the data of both cases and controls were collected according to a defined protocol and were electronically stored in a dedicated database. Finally, the small number of cases complicated by FGR in the study group limits the interpretation of the data on the laterality of the missing artery in SUA fetuses.

## 5. Conclusions 

Our study confirms that fetuses presenting with an isolated SUA at mid-pregnancy are at increased risk of abnormal fetal growth and shows that the subgroup of pregnancies subsequently complicated by FGR presents with a smaller diameter of the remaining umbilical arteries at 20–22 weeks. This may be due to a failure of the remaining artery to compensate hemodynamically and thus could be used to identify fetuses at risk of FGR later in pregnancy.

## Figures and Tables

**Figure 1 diagnostics-13-00571-f001:**
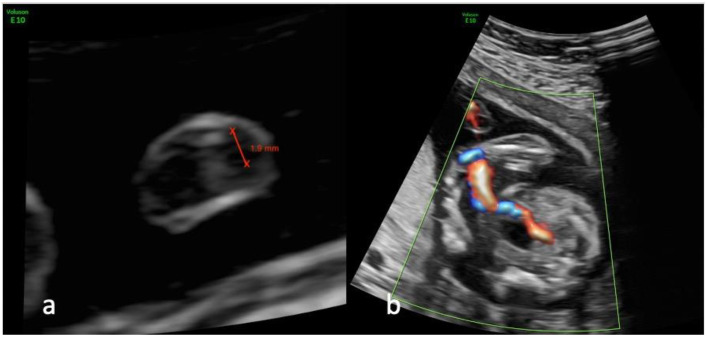
Ultrasound views of the umbilical cord and its insertion at the level of the bladder in a fetus presenting a single umbilical artery at 20 weeks of gestation; (**a**) transverse view of a free loop of umbilical cord showing the measurement of the remaining artery; (**b**) color Doppler imaging mapping the insertion of the umbilical cord at the level of the bladder.

**Figure 2 diagnostics-13-00571-f002:**
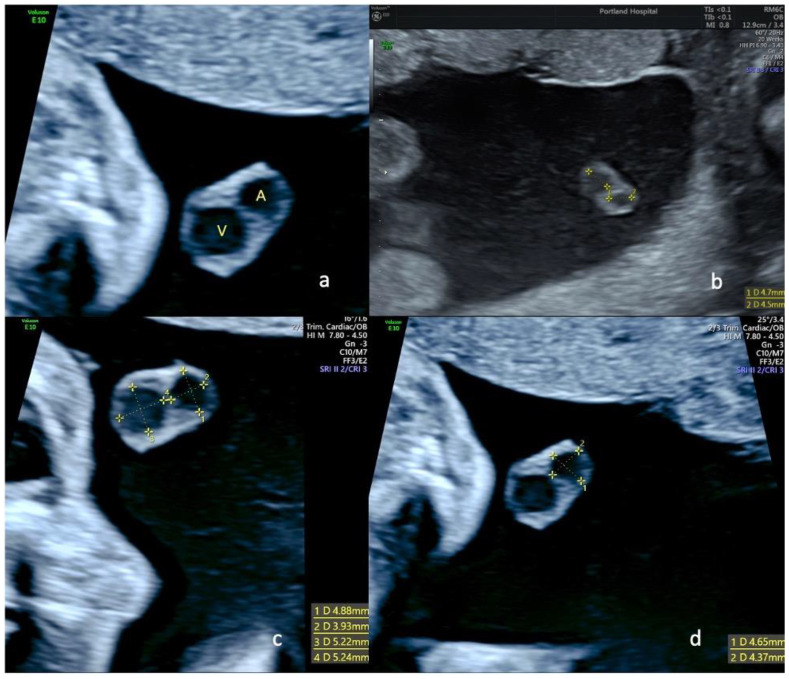
Ultrasound views of the free portion of the umbilical cord in a fetus with single umbilical artery at 22 weeks of gestation; (**a**) transverse view of the free loop showing the remaining artery (A) and the umbilical vein (V); in pictures (**b**–**d**), we can see different measurements of arteries and veins in a transverse view of a free loop of umbilical cord at 22 weeks in fetuses with SUA.

**Figure 3 diagnostics-13-00571-f003:**
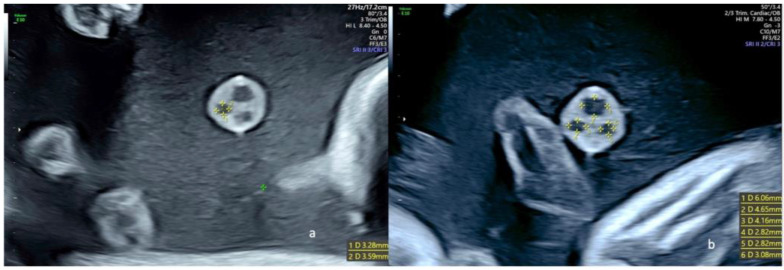
Ultrasound views of a free umbilical cord in fetuses with a regular umbilical cord at 22 weeks of gestation; (**a**) one of the two umbilical arteries was measured in a transverse view of a free loop, and the diameters were 3.28 × 3.59 mm; (**b**) transverse view of a free loop showing the measure of both umbilical arteries (4.16 × 2.82 and 2.82 × 3.08 mm) and the umbilical vein (6.06 × 4.65 mm).

**Figure 4 diagnostics-13-00571-f004:**
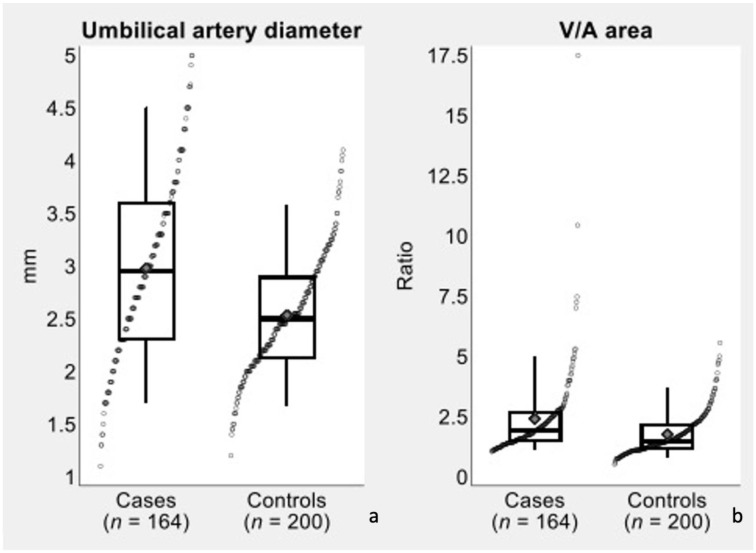
Dot plot combined with box plot showing the diameter of umbilical artery (**a**) and the vein-to-artery (V/A) area ratio (**b**), in cases (single artery) versus controls (two arteries). In the control group, diameter estimates were obtained as the average of the diameters of the two umbilical arteries, while area estimates contributing to the V/A ratios were obtained as the sum of the areas of the two umbilical arteries. The following is an explanation of the elements that constitute a box plot: the median is shown by the line that divides the box into two parts; the range of scores from the lower to upper quartile is marked by the box; whiskers stretch from the 5th percentile to the 95th percentile; the gray diamond inside the plot represents the mean of the data.

**Table 1 diagnostics-13-00571-t001:** Demographics and clinical characteristics of the study sample, overall and by normal, SGA and FGR status.

Characteristic	All	Normal	SGA	FGR	*p* Value
	(*n* = 164)	(*n* = 137)	(*n* = 8)	(*n* = 19)	
Maternal age, y					0.791
≤30	55 (33.5%)	45 (32.8%)	3 (37.5%)	7 (36.8%)	
31–35	52 (31.7%)	43 (31.4%)	3 (37.5%)	6 (31.6%)	
36–40	49 (29.9%)	42 (30.7%)	1 (12.5%)	6 (31.6%)	
>40	8 (4.9%)	7 (5.1%)	1 (12.5%)	0 (0.0%)	
Ethnicity					0.943
White	148 (90.2%)	123 (89.8%)	8 (100.0%)	17 (89.5%)	
Black	12 (7.3%)	10 (7.3%)	0 (0.0%)	2 (10.5%)	
Asian	3 (1.8%)	3 (2.2%)	0 (0.0%)	0 (0.0%)	
Unknown	1 (0.6%)	1 (0.7%)	0 (0.0%)	0 (0.0%)	
Body mass index, kg/m²					0.674
<25	59 (36.0%)	47 (34.3%)	4 (50.0%)	8 (42.1%)	
25–29.9	59 (36.0%)	53 (38.7%)	2 (25.0%)	4 (21.1%)	
≥30	20 (12.2%)	17 (12.4%)	1 (12.5%)	2 (10.5%)	
N/A	26 (15.9%)	20 (14.6%)	1 (12.5%)	5 (26.3%)	
Parity					0.600
0	91 (55.5%)	76 (55.5%)	5 (62.5%)	10 (52.6%)	
1	45 (27.4%)	37 (27.0%)	2 (25.0%)	6 (31.6%)	
≥2	14 (8.5%)	14 (10.2%)	0 (0.0%)	0 (0.0%)	
N/A	14 (8.5%)	10 (7.3%)	1 (12.5%)	3 (15.8%)	
Side of remaining umbilical artery					0.226
Right	100 (61.0%)	80 (58.4%)	5 (62.5%)	15 (78.9%)	
Left	64 (39.0%)	57 (41.6%)	3 (37.5%)	4 (21.1%)	
Umbilical artery RI					0.707
≤0.6	33 (20.1%)	27 (19.7%)	1 (12.5%)	5 (26.3%)	
>0.6	131 (79.9%)	110 (80.3%)	7 (87.5%)	14 (73.7%)	
Mean RI of uterine arteries					0.160
≤0.5	33 (20.1%)	30 (21.9%)	2 (25.0%)	1 (5.3%)	
>0.5	96 (58.5%)	75 (54.7%)	5 (62.5%)	16 (84.2%)	
N/A	35 (21.3%)	32 (23.4%)	1 (12.5%)	2 (10.5%)	
Mean PI of uterine arteries					0.195
≤1	55 (33.5%)	44 (32.1%)	4 (50.0%)	7 (36.8%)	
>1	37 (22.6%)	30 (21.9%)	0 (0.0%)	7 (36.8%)	
N/A	72 (43.9%)	63 (46.0%)	4 (50.0%)	5 (26.3%)	
Gestational age at delivery, wk					0.016 *
<39	87 (53.0%)	66 (48.2%)	7 (87.5%)	14 (73.7%)	
≥39	77 (47.0%)	71 (51.8%)	1 (12.5%)	5 (26.3%)	
Fetal sex					0.252
Male	89 (54.3%)	77(56.2%)	3 (37.5%)	9 (47.3%)	
Female	75 (45.7%)	60 (43.8%)	5 (62.5%)	10(52.6%)	
Birthweight percentile					<0.001 *
<10°	33 (20.1%)	17 (12.4%)	3 (37.5%)	13 (68.4%)	
10–90°	120 (73.2%)	109 (79.6%)	5 (62.5%)	6 (31.6%)	
>90°	11 (6.7%)	11 (8.0%)	0 (0.0%)	0 (0.0%)	

* *p* value ≤ 0.05; Abbreviations: SGA, small for gestational age; FGR, fetal growth restriction; RI, resistive index; PI, pulsatility index.

**Table 2 diagnostics-13-00571-t002:** Results of logistic regression and nonparametric receiver operating characteristic (ROC) analysis conducted on umbilical artery diameter and vein-to-artery (V/A) area ratio to identify small-for-gestational-age fetuses and cases of fetal growth restriction.

	Artery Diameter, mm		V/A Area Ratio	
	Estimate	95% CI	Estimate	95% CI
Small for gestational age				
Odds ratio	0.72	0.31–1.68	1.05	0.76–1.46
Area under ROC curve, %	57	35–80	59	38–81
Optimal cutoff	3.3	1.4–4.4	1.94	1.32–4.27
Sensitivity at cutoff, %	88	53–98	75	41–93
Specificity at cutoff, %	33	26–41	54	46–61
Accuracy * at cutoff, %	36	29–44	55	47–62
Fetal growth restriction				
Odds ratio	0.60	0.33–1.09	1.09	0.88–1.35
Area under ROC curve, %	62	48–76	51	36–67
Optimal cutoff	3.1	1.9–3.7	1.72	1.17–2.42
Sensitivity at cutoff, %	79	57–91	53	32–73
Specificity at cutoff, %	44	36–52	63	55–71
Accuracy * at cutoff, %	48	41–56	62	55–69

* True positives plus true negatives over *n*; Abbreviations: CI, confidence interval.

## Data Availability

The data presented in this study are available on request from the corresponding author. The data are not publicly available due to privacy.

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
