# Peer review of "Changes in Artery Diameters and Fetal Growth in Cases of Isolated Single Umbilical Artery"

_diagnostics, 2023, doi:10.3390/diagnostics13030571_

Round 1

Reviewer 1 Report

It is an interesting study, I enjoyed reading it.

Research is a dynamic activity and it is gradually growing but not reaching maturity. However, I have some observations but my observations are not judgmental and can be challenged and subjected to override if seemed unimportant to the author or Editor. I’ll feel pleasure if I was contacted for that.

1-       Title: The title of the study is looking to me a bit deficient “Changes in artery diameters and fetal growth in cases of isolated single umbilical artery”.

In my opinion, it is a comparison of fetal growth and isolated SUA. Moreover, the comparison has been made sonographically. Therefore if the title is rephrased as “Sonographic comparison of fetal growth and isolated single umbilical artery diameter”.

2-       Study Design: The respected authors wrote the study design as “retrospective longitudinal cohort study”

By retrospective, we mean that the sample is followed backward in time. For example, a patient is examined and his previous record is checked as history and used as a variable in the study. But the patients are followed in the current study therefore it is a longitudinal prospective study.

Cohort study is a term used for a controlled population, particularly in prospective longitudinal studies but the patients coming for ultrasound examination couldn’t be a controlled population. So, in my opinion, the word cohort is not suitable for this study. I think the study design should be “Prospective longitudinal observational study”.

3-       Patient selection: In polyhydramnios and normal AFI it is easy to focus the umbilical cord and measure the UA diameter but in the cases of Oligohydramnios, practically it is difficult to measure the UA diameter. However, if there were no patients of oligohydramnios in the sample. Then I suggest the author to add oligohydramnios in the exclusion section.

4-       In the section “Patients and ultrasound examination” second paragraph, a sentence is written, which is a bit confusing “confirmed by measurements of the fetal crown-rump length (CRL) at the first-trimester dating scan and the 11-14 weeks ultrasound examination.” I think the entire period in which CRL is taken should be expressed in Weeks instead of trimesters to eliminate confusion. I suggest “confirmed by the sonographic measurement of the fetal crown-rump length (CRL) from 7-14 weeks”.

5-       Statistical analysis: In the statistical analysis section three study groups are taken by the authors, “normal growth (10-90th centile), SGA (birth weight of < than 10th percentile for gestational age) and FGR (estimated fetal weight or abdominal circumference that is <than the 10th percentile for gestational age)”

Here the differentiation between SGA and FGR is a bit difficult. If the author disagrees then let it be the same.

6-       Remaining umbilical artery is used several times, I think, the single umbilical artery is enough to be used instead of remaining umbilical artery. If the author disagrees then let it be the same.

7-       If means of the PI, RI, and S/D ratios are compared in SUA and DUA cords then it will yield too meaningful results. If the authors do it in the other paper with the title “Comparison of umbilical artery Doppler indices in the single and double umbilical artery cords”.

8-       Figure 4: Both sides of the figures should be described separately while giving them names as A and B or left and right sides.

Author Response

Dear Editors, please find attached a revised version of our manuscript: Changes in artery diameters and fetal growth in cases of isolated single umbilical artery”. The manuscript has been modified according to reviewers’ suggestions, with changes in the text highlighted in yellow. An itemized response to reviewers’ comments is also included.

Reviewer #1:

It is an interesting study, I enjoyed reading it.

Research is a dynamic activity and it is gradually growing but not reaching maturity. However, I have some observations but my observations are not judgmental and can be challenged and subjected to override if seemed unimportant to the author or Editor. I’ll feel pleasure if I was contacted for that.

1-       Title: The title of the study is looking to me a bit deficient “Changes in artery diameters and fetal growth in cases of isolated single umbilical artery”.

In my opinion, it is a comparison of fetal growth and isolated SUA. Moreover, the comparison has been made sonographically. Therefore if the title is rephrased as “Sonographic comparison of fetal growth and isolated single umbilical artery diameter”.

Reply: Thank you. We prefer to keep the title “Changes in artery diameters and fetal growth in cases of isolated single umbilical artery”.

2-       Study Design: The respected authors wrote the study design as “retrospective longitudinal cohort study”

By retrospective, we mean that the sample is followed backward in time. For example, a patient is examined and his previous record is checked as history and used as a variable in the study. But the patients are followed in the current study therefore it is a longitudinal prospective study.

Cohort study is a term used for a controlled population, particularly in prospective longitudinal studies but the patients coming for ultrasound examination couldn’t be a controlled population. So, in my opinion, the word cohort is not suitable for this study. I think the study design should be “Prospective longitudinal observational study”.

Reply: Thank you for your comment. We modified the test according to your suggestion.

3-       Patient selection: In polyhydramnios and normal AFI it is easy to focus the umbilical cord and measure the UA diameter but in the cases of Oligohydramnios, practically it is difficult to measure the UA diameter. However, if there were no patients of oligohydramnios in the sample. Then I suggest the author to add oligohydramnios in the exclusion section.

Reply: The population of the study included uncomplicated pregnancies in non-smoking mothers with a singleton fetus with no anomalies on ultrasound examination confirmed at birth collected during the same study period. We excluded multiple pregnancies, fetuses presenting aneuploidy or other congenital anomalies and cases with active maternal smoking were excluded from the study group. In our study simple we didn’t find difficult in the measure of the umbilical vessel’s diameters maybe because of the rarity of oligohydramnios in fetuses at 20 weeks with no anomalies on ultrasound examination.

4-       In the section “Patients and ultrasound examination” second paragraph, a sentence is written, which is a bit confusing “confirmed by measurements of the fetal crown-rump length (CRL) at the first-trimester dating scan and the 11-14 weeks ultrasound examination.” I think the entire period in which CRL is taken should be expressed in Weeks instead of trimesters to eliminate confusion. I suggest “confirmed by the sonographic measurement of the fetal crown-rump length (CRL) from 7-14 weeks”.

Reply: Thank you for your comment. We modified the test according to your suggestion.

5-       Statistical analysis: In the statistical analysis section three study groups are taken by the authors, “normal growth (10-90th centile), SGA (birth weight of < than 10th percentile for gestational age) and FGR (estimated fetal weight or abdominal circumference that is <than the 10th percentile for gestational age)”

Here the differentiation between SGA and FGR is a bit difficult. If the author disagrees then let it be the same.

Reply: Thank you for your comment. We prefer to hold the differentiation between SGA and FGR because in the clinical practice SGA and FGR are two identities so different with different management.

6-       Remaining umbilical artery is used several times, I think, the single umbilical artery is enough to be used instead of remaining umbilical artery. If the author disagrees then let it be the same.

Reply: Thank you for your comment. We modified the test according to your suggestion.

7-       If means of the PI, RI, and S/D ratios are compared in SUA and DUA cords then it will yield too meaningful results. If the authors do it in the other paper with the title “Comparison of umbilical artery Doppler indices in the single and double umbilical artery cords”.

Reply: Thank you for your comment. We previously did in the article “Reference charts for umbilical Doppler pulsatility index in fetuses with isolated two-vessel cord”.

8-       Figure 4: Both sides of the figures should be described separately while giving them names as A and B or left and right sides.

Reply: Thank you for your comment. We modified the figure 4 according to your suggestion.

Reviewer 2 Report

Thank you for inviting me to review this article addressing a very challenging topic, equally for clinical research and practicing physicians. There are several recent studies and meta-analyses suggesting that an isolated single umbilical artery (SUA) is significant related to the risk of adverse perinatal outcome and labor complications.

The manuscript is original and of scientific interest for the involved clinicians, taking into account both the scarce information in this field and the increasing incidence of this anomaly, even isolated, as authors also demonstrate in the introduction.

A strength of the article is that current information complete the previous own study (mentioned by authors as reference 13), providing similar data with the most recent Li’s study (reference 20).

The current study has a clear and innovative objective, the data are updated as reflected in the selected references, and results are clearly illustrated by tables and figures. I especially appreciate the section of discussion which is well organized, compares the own findings with the most recent mentioned study and also highlights the clinical and prognostic importance of the problem.

Another obvious strength is that the study adds a new parameter for monitoring abnormal fetal growth, as no “studies have evaluated the changes in fetal growth in relation with the change in the diameter of the remaining artery”. Thus, the authors propose this parameter as a more simple measurement besides Doppler data, which “could be used to identify fetuses at risk of abnormal growth later in pregnancy”. Further studies are needed to demonstrate the right significance of the cut-off value for the diameter of the remaining umbilical artery, as authors suggest.

On the other hand, the retrospective nature of the study, conducted in two different centers, design which can affect the interpretation of ultrasound measurements, and also the small number of patients included, represent the limitations of this research. In fact, all results in literature are based on retrospective data excepting Li’s recent prospective study. So, it would be useful to initiate a prospective multicenter trial to validate and complete the data.

Author Response

Dear Editors, please find attached a revised version of our manuscript: Changes in artery diameters and fetal growth in cases of isolated single umbilical artery”. The manuscript has been modified according to reviewers’ suggestions, with changes in the text highlighted in yellow. An itemized response to reviewers’ comments is also included.

Reviewer #2:

Thank you for inviting me to review this article addressing a very challenging topic, equally for clinical research and practicing physicians. There are several recent studies and meta-analyses suggesting that an isolated single umbilical artery (SUA) is significant related to the risk of adverse perinatal outcome and labor complications.

The manuscript is original and of scientific interest for the involved clinicians, taking into account both the scarce information in this field and the increasing incidence of this anomaly, even isolated, as authors also demonstrate in the introduction.

A strength of the article is that current information complete the previous own study (mentioned by authors as reference 13), providing similar data with the most recent Li’s study (reference 20).

The current study has a clear and innovative objective, the data are updated as reflected in the selected references, and results are clearly illustrated by tables and figures. I especially appreciate the section of discussion which is well organized, compares the own findings with the most recent mentioned study and also highlights the clinical and prognostic importance of the problem.

Another obvious strength is that the study adds a new parameter for monitoring abnormal fetal growth, as no “studies have evaluated the changes in fetal growth in relation with the change in the diameter of the remaining artery”. Thus, the authors propose this parameter as a simpler measurement besides Doppler data, which “could be used to identify fetuses at risk of abnormal growth later in pregnancy”. Further studies are needed to demonstrate the right significance of the cut-off value for the diameter of the remaining umbilical artery, as authors suggest.

On the other hand, the retrospective nature of the study, conducted in two different centers, design which can affect the interpretation of ultrasound measurements, and also the small number of patients included, represent the limitations of this research. In fact, all results in literature are based on retrospective data excepting Li’s recent prospective study. So, it would be useful to initiate a prospective multicenter trial to validate and complete the data.

Reply: Thank you. We greatly appreciate your comments.
